# Precarious Young Adults’ Mental Health during the Pandemic: The Major Impact of Food Insecurity Independently of COVID-19 Diagnosis

**DOI:** 10.3390/nu15143260

**Published:** 2023-07-24

**Authors:** Aziz Essadek, Gérard Shadili, Pablo Bergami Goulart Barbosa, Adèle Assous, Frédéric Widart, Ségolène Payan, Thomas Rabeyron, Emmanuelle Corruble, Bruno Falissard, Florence Gressier

**Affiliations:** 1INTERPSY Laboratory, Faculty of Psychology, University of Lorraine, 54015 Nancy, France; thomas.rabeyron@univ-lorraine.fr; 2PCPP Laboratory, Institute of Psychology, University of Paris Cité, 75006 Paris, France; gerard.shadili@imm.fr; 3Institut Mutualiste Montsouris, 75014 Paris, France; pbergami@gmail.com; 4CRPMS Laboratory, Faculty of Psychology, University of Paris Cité, 75006 Paris, France; adeleassous@hotmail.fr; 5Faculty of Psychology, University of Liège, 4000 Liège, Belgium; fredericwidart@ik.me; 6Essonne Research and Social Work Institute, IRFASE, 91034 Evry, France; segolenepayan@gmail.com; 7Department of Psychiatry, Bicêtre University Hospital, Assistance Publique Hôpitaux de Paris APHP, Hôpitaux Universitaires Paris Saclay, 94275 Le Kremlin Bicêtre, France; emmanuelle.corruble@aphp.fr (E.C.); florence.gressier@aphp.fr (F.G.); 8CESP, INSERM U1018, Moods Team, Faculté de Médecine Paris Saclay, University Paris-Saclay, 94275 Le Kremlin Bicêtre, France; 9Department of Public Health, School of Medecine, University Paris-Saclay, 94270 Le Kremlin Bicêtre, France; bruno.falissard@universite-paris-saclay.fr; 10CESP, INSERM 1018, University Paris-Saclay, 94275 Le Kremlin Bicêtre, France

**Keywords:** food insecurity, mental health, suicidal ideation, COVID-19, precarious, young adults

## Abstract

The COVID-19 pandemic had a major impact on mental health across populations, especially young and precarious people. Furthermore, COVID-19 diagnosis itself has been associated with psychiatric symptoms. However, only a few studies have assessed the mental health of precarious youth, and examined a possible association with food insecurity, while including COVID-19 diagnosis in their analyses. We aimed to determine the prevalence of poor mental health in precarious youth during the COVID-19 pandemic, and to investigate its possible association with food insecurity, independently of COVID-19 diagnosis. In a cross-sectional study conducted in the context of an employment program for precarious youth (18–25 years) living in Paris, France, 823 individuals were assessed for depression, anxiety, subjective distress and food insecurity during the second lockdown of 2020. A directed acyclic graph (DAG)-based approach was used to identify confounders for inclusion in a multivariate regression model. Of the 823 precarious youth, 45.93% reported significant symptoms of depression, 36.69% anxiety, 39% distress and 25.39% suicidal ideation. In the multivariate analysis based on DAG, food insecurity (less than one meal per day) was associated with depression (OR = 2.30; CI%: 1.19–4.51), anxiety (OR = 2.51; CI%: 1.29–4.88), distress (OR = 2.36; CI%: 1.23–4.57) and suicidal ideation (OR = 4.81; CI%: 2.46–9.44), independently of age, gender, education, COVID-19 contact and COVID-19 diagnosis. This study highlights the importance of food insecurity on mental health among young precarious people during the COVID-19 pandemic. Reducing food insecurity is essential to help reduce psychological distress.

## 1. Introduction

The COVID-19 pandemic led to an economic crisis and accentuated pre-existing social inequity. It also increased the prevalence of mental health issues in all populations, and especially among those living in precarious conditions [1,2]. Precariousness is defined as “the absence of one or more of the securities enabling individuals and families to assume their basic responsibilities and enjoy their fundamental rights” [3]. For example, precariousness is related to a lack of permanent housing, a lack of permanent employment, and insufficient social and interpersonal support. Across age groups, precariousness seemed to be associated with poor mental health during the pandemic. One study reported that around 30% of the homeless and those living in shelters presented moderate to severe symptoms of depression [4].

Young adults have been identified as the age group most exposed to psychological distress during the COVID-19 pandemic [5,6]. Psychological vulnerability seems particularly high among the young adult population living in precarious conditions [7].

Furthermore, young students facing financial uncertainty or insecurity were reported to have more symptoms of anxiety and depression [8], as well as suicidal ideation [9]. The mental health of students living alone has also been shown to be significantly impacted by the pandemic [10]. These findings concur with the results of previous studies, highlighting the particular vulnerability of young people in precarious situations to psychological distress [11,12].

Recent studies also showed that the risk of being in a situation of food insecurity (that is, inadequate access to sufficient, safe and nutritious food that meets individuals’ dietary needs [13]) was increased by approximately 30% during the COVID-19 pandemic, mainly among people in precarious situations, which had a major impact on psychological distress [14,15]. A recent study found that food insecurity at the time of the first lockdown still affected eating behaviors 6 months later [16]. In addition, COVID-19 disease itself is associated with poor mental health. Neurological and neuropsychiatric disorders were reported in young adults after a COVID-19 diagnosis [17]. Depression and anxiety could be a direct symptom of the disease [18] or related to its long-term effects [19]. Finally, individuals are also impacted when someone close to them contracts COVID-19, either because they are worried about that person or because they fear contracting the disease [20]. Young adults in precarious situations are, therefore, particularly exposed to psychological distress, regardless of a previous diagnosis of COVID-19, even though such a diagnosis could cause further deteriorations in their mental health.

However, few studies have specifically addressed the mental health of precarious youth during the pandemic. Indeed, the French literature has shown little interest in the issue of food insecurity during the pandemic. A study focusing on the homeless indicated the importance of food distress on their mental health [4], and a study on the general population also found a reduction in the PNNS-GS2 index among students experiencing financial precariousness, food insecurity, and psychological distress [21]. However, we have not identified any studies on this topic among a population of young precarious individuals who are not enrolled in university. Consequently, it can be affirmed that there is a lack of research on this population. The aim of this study was: (1) to assess mental health in precarious young people during the second wave of the COVID-19 pandemic; (2) to explore if food insecurity was a risk factor for mental health in this population, independently of COVID-19 diagnosis.

We hypothesized that food insecurity was predictive of high psychological distress.

## 2. Methods

### 2.1. Data Source

Our study examines the mental health of precarious young people participating in a labor market inclusion program (Mission Locale de Paris) led by the city of Paris in France. This socio-professional inclusion program welcomes young people aged from 16 to 25, and works primarily with the populations described above, who were the most vulnerable to the negative effects of the pandemic on mental health and seeking employment in the year 2020.

A text message was sent to the cell phones of 10,000 youth registered at the Mission Locale who had met with one of its counselors in 2020. The recipients of the message were over 18 years of age. The message directed them to the Lime Survey software, on which the study was hosted. In addition, printed questionnaires were available in the reception centers of the program. All participants provided their on-line informed consent.

This phase was carried out between 9 December and 14 December 2020, one week after the peak of the epidemic during the second lockdown in France.

### 2.2. Measures

Questionnaires included socio-demographic data: age; gender; level of education; type of housing; employment status; access to state financial support; their level of income (no income, <€500, between €500 and €1000, >€1000). In addition, they were asked if they had contracted COVID-19 (“Not aware”, “I was diagnosed by PCR test and/or a doctor”), and whether they had been in contact with a person infected by COVID-19. An indicator of food insecurity was assessed by the number of meals they could afford in a day. The question was asked in the following way: since the beginning of the lockdown, on average, you were able to afford (“2 or more meals a day”, “1 meal a day” and “less than 1 meal a day”).

Participants were asked to fill out three mental health scales, validated in French, to measure levels of depression (Patient Health Questionnaire, PHQ-9; range 0–27;) [22], anxiety (Generalized Anxiety Disorder, GAD-7; range 0–21) [23] and subjective distress (Impact of Event Scale-Revised, IES-R; range 0–88) [24]. The PHQ-9 and GAD-7 were chosen as they are widely cited tools for symptom detection in primary care. The cut-off was determined for depression scores “10” [25], anxiety “8” [26] and distress “26” [27]. The IES-R is a self-administered questionnaire that is frequently used to assess subjective distress caused by traumatic events [28]. To determine the presence of suicidal ideation, we used item 9 of PHQ-9: “Over the last two weeks, how often have you been bothered by thoughts that you would be better off dead, or thoughts of hurting yourself in some way”. The possible answers are “not at all”, “several days”, “more than half the days” and “almost every day”. Suicidal ideation was defined as “present” if respondents checked “several days”, “more than half the days” and “almost every day”. It was considered “absent” if the answer was “not at all” [29].

Questionnaires were anonymous to ensure the confidentiality and reliability of the data. This study has been approved by the University of Lorraine and is registered under the number 2021–142.

### 2.3. Statistical Analysis

Statistical analysis was carried out using the R 4.0.2 software. First, a descriptive analysis was carried out. Quantitative variables were expressed as means and standard deviations (SD) and qualitative variables by percentages.

Then, simple logistic regressions were used to determine which variables were associated with high levels of depression, anxiety, distress, and suicidal thoughts.

Pearson’s correlation, Phi coefficients, and variation inflation factors (VIF < 10) were calculated to check for non-correlation and multicollinearity [30].

To specifically explore whether food insecurity was an independent risk factor regarding the mental health of precarious youth, we used a directed acyclic graph (DAG)-based approach (Figure 1) to select the variables to include in the regression model [31]. We conceptualized age, gender, education, COVID-19 contact and COVID-19 diagnosis as potential confounders, as these variables can be a common cause of both the exposure and the outcome. We conceptualized the type of housing, employment status, and access to state financial support as possible ancestors to exposure. Ancestors to exposure were not adjusted in the regression model since adjustment could bias the analysis. The DAG was created using DAGitty version 2.3.28. All tests were two-tailed. The significance level was set at 0.05.

## 3. Results

### 3.1. Characteristics of the Sample

Our study included a sample of 823 precarious youths. The sample characteristics are described in Table 1.

About half of the participants, 410 (49.82%), were women. The mean age of the population was 21.23 (SD = 2.11). In this study, 239 (29.04%) participants did not have any educational degree and 426 (51.77%) had a diploma below the *Baccalauréat* (secondary education completion diploma). Two hundred and fifty-seven (31.23%) participants lived alone. Six hundred and forty-five (78.37%) participants reported being unemployed and 195 (23.69%) reported receiving state financial assistance. Among all participants, 437 (53.10%) lived with their parents, 111 (13.49%) lived in shelters or social housing, and 66 (8.02%) were homeless. Four hundred and thirteen (50.18%) had no income, 220 (26.73%) had income below €500, 131 had an income between €500 and €1000, and 59 (7.17%) had an income above €1000. Regarding food insecurity, 429 (52.13%) participants ate two or more meals per day, 353 (42.89%) ate one meal per day, and 41 (4.98%) ate less than one meal per day. Also, 56 (6.81%) were diagnosed with COVID-19.

### 3.2. Mental Health

The average score on the depression scale was 9.35 [SD = 2.11], for the anxiety scale it was 6.37 [SD = 5.96] and the average distress score was 22.35 [SD = 5.96]. After applying cut-offs, 45.93% of participants reported significant symptoms of depression, 36.69% significant symptoms of anxiety, 39% significant symptoms of distress and 25.39% suicidal ideation.

### 3.3. Factors Associated with Depression, Anxiety, Distress and Suicidal Ideation

No correlation was identified between the explanatory variables (all <0.4). Also, the analysis of VIF indicated an absence of multicollinearity (<1.3).

The results of the bivariate for depression, anxiety, distress and suicidal ideation are presented in Table 2.

To summarize, depression was associated with gender [Female OR = 1.39 (1.05–1.83); *p* = 0.02], COVID-19 diagnosis, COVID-19 contact and food insecurity. Anxiety was associated with gender, COVID-19 diagnosis, COVID-19 contact and food insecurity. Also, educational level lower than a high school education was negatively associated with anxiety. Distress was associated with receiving financial assistance, living alone, being unemployed, having had contact with COVID-19, having a COVID-19 diagnosis, and food insecurity. Suicidal ideation was significantly associated with living in social housing and mainly associated with being homeless, living on less than 500 euros per month, living alone, and food insecurity.

In the multivariate analysis based on DAG, food insecurity was associated with depression, anxiety, distress and suicidal ideation, independently of age, gender, education, COVID-19 contact and COVID-19 diagnosis (Table 3).

Odds ratios represent the risk of depressive symptoms, anxiety, stress, or suicidal ideation as a function of the level of dietary distress. The model was adjusted for age, gender, education level, being in contact with a COVID+ person, and having contracted COVID-19.

## 4. Discussion

### 4.1. Prevalence

Our study reported that 45.93% of precarious young people showed significant symptoms of depression, 36.69% had significant symptoms of anxiety, 39.00% had distress symptoms and 25.39% experienced suicidal ideation.

These high rates of psychological distress are above those of people of the same age group during the first lockdown [32]. They are also higher than those found by Scarlett et al. [4] in a study conducted with the homeless. This finding is consistent with the analysis of Winkler et al. [33], who suggest that young adults and students were more prone to deteriorating mental health during the second lockdown. These elements also concur with other research, which suggests that pandemics and disasters have a major impact on poor and vulnerable populations. This effect may be partially explained by the significant increase in economic inequality and in the disparities in access to prevention and primary care these periods create [2].

### 4.2. Food Insecurity

In our study, the proportion of participants who had access to one meal (353; 42.89%) and less than one meal (41; 4.98%) per day is important. Our results suggest that the less these youth can eat, the more they are exposed to high levels of psychological suffering. The associations remained significant after adjustment. Our results are consistent with pre-pandemic studies that established a link between depression and food insecurity [34]; however, they mainly focused on cohorts of homeless people. Our study underlines that this association can be extended to other forms of precariousness and that the dimension of food insecurity is a central factor. Despite the different measures used and a focus on the homeless, the study conducted by Scarlett et al. [4] also highlighted an association between food distress and depression, which was reported in other studies targeting other populations [16]. Previous studies have brought attention to the bidirectional association between depression and food insecurity [35,36] and explored the possibility of mutually reinforcing cycles between food insecurity and poor mental health [37]. It is also important to highlight that food insecurity leads to malnutrition, and that malnutrition can have detrimental effects on mental health [38]. Studies on this population have underlined the coexistence of psychological distress and food insecurity [37,39]. Our study complements this literature by focusing on precarious youth and highlighting the importance of food insecurity in the development of psychological troubles, even during a pandemic. Similar results have been found in early care and education (ECE) workers, a vulnerable population often working in precarious conditions [40]. Indeed, our results suggest that, despite the impact of the pandemic and lockdowns on mental health, food insecurity is an important factor, especially regarding suicidal ideation. Our results are also consistent with the few studies that focus on the association between suicidal ideation and food distress [39]. An Australian study [41] points out that the feeling of hopelessness was more present among precarious young people during this pandemic period, and an American study highlighted the importance of hopelessness among undernourished youth [42]. Thus, fear of the future associated with food insecurity may explain the high proportion of suicidal thoughts in this sample of precarious young people.

Considering the link between food insecurity and psychological distress is all the more important in this pandemic period, when food insecurity is on the rise [43,44] especially within the increasingly precarious student population, where its incidence rate is up to 34.5% [45]. As Loftus et al. [44] point out, governments are taking notice, and they are providing food aid. It would, therefore, be relevant to assess the proportion to which populations most in need of this aid actually benefit from it. This question is paramount because food distress is often the source of shame [42] and social stigma [46], which are disincentives for people to access food assistance. This context may also lead to changes in eating behaviors that promote weight gain and degrade body health [47]. It would also be necessary to assess whether food aid responds proportionately to the needs of these groups. Indeed, reducing the risk of exposure to food insecurity and allowing for the most precarious populations (and notably the youngest part of those populations) to have a suitable daily food intake seems to significantly reduce their vulnerability to psychological suffering and suicidal thoughts, and could have an impact on eating behaviors in future years [48].

### 4.3. Limits

The results of our study have some limitations, which should be pointed out. First, our cross-sectional survey does not allow for conclusions to be drawn on the evolution of the situation. Longitudinal research would produce more data indicating the possible direction that the detected associations may take over time. It could also shed light on the bi-directionality between precariousness and mental health. Secondly, as the survey is based on self-assessment scales, there is no ruling out the possibility that individuals responded in a way that positively influenced their results. Thirdly, there may be an under-representation of depressed people who were unwilling to respond as a consequence of their pathological state, despite having participated in the social program in 2020, which may have consequences for the statistical power of the study. Also, our sample size is small, and represents only 8.23% of program participants. It should also be noted that there could be an overlap between the notion of “food insecurity” and the presence of “eating disorders’’, since the latter are often associated with depression. However, the research questionnaire addressed access to food and not the desire for it, and this distinction seems to be confirmed by statistical analysis, which shows that the item concerning eating disorders in the PHQ-9 were not correlated with the assessment of food insecurity. Also, the way that food insecurity is assessed is a limitation. The use of a scale such as the Food Insecurity Experience Scale would have provided more standardized results. A final limitation of our study is the lack of an ethnic component in our sociodemographic data. However, that is due to the fact that French law forbids collecting this sort of data.

## 5. Conclusions

Despite these limitations, our data shed light on one of the most important aspects of this pandemic: its psychological and social impacts. The high incidence rates of depression, anxiety and especially suicidal ideation in the study sample underline that the influence of the current period on the mental health of young people should not be underestimated. It should be one of the key factors taken into consideration by policymakers when devising measures to control the pandemic an curb the economic crisis that has resulted from it. Likewise, the strong association between food insecurity and suicidal ideation underlines the importance of considering the mental well-being of these populations holistically, and not neglecting the importance of the fundamental elements of existence concerning organic security. This stresses the necessity of addressing the matter through political and collective action, establishing more food aid programs for precarious populations and properly communicating on the matter to ensure the concerned population is aware of the accessibility of such aid. It also seems clear that the associations explored in this study should inform the clinical interventions of psychologists and psychiatrists working with precarious youth. Our study opens up prospects for expansion (longitudinal study and enlargement of the sample), which seem interesting to pursue in order to produce even more solid data concerning the mental health of precarious young people and its evolution during this time of great social upheaval. Finally, our study highlights the role that food insecurity plays in the development of psychological suffering, and particularly in the construction of suicidal ideas among people in precarious situations. Further studies are needed on this issue beyond the pandemic period.

## Figures and Tables

**Figure 1 nutrients-15-03260-f001:**
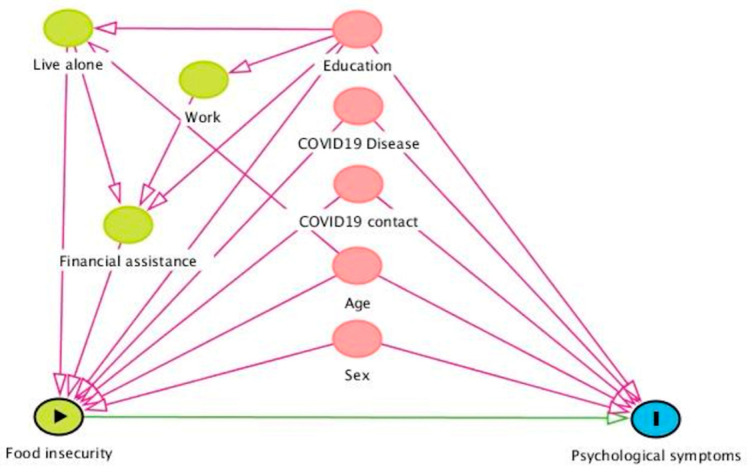
Directed acyclic graph representing the causal assumptions used for covariate selection.

**Table 1 nutrients-15-03260-t001:** Characteristics of the sample.

Characteristic	Total *n* = 823 n (%) or Mean [SD]
Age	21.23 [2.11]
Gender	
Female	410 (49.82)
Level	
Higher than baccalaureate	116 (14.10)
Baccalaureate	281 (34.14)
Lower than baccalaureate	426 (51.76)
Accommodation	
Family	514 (62.5)
Tenant	178 (21.6)
Emergency shelter	65 (7.9)
Homeless	66 (8)
Live Alone	
Yes	257 (31.23)
Financial assistance	
Yes	195 (23.69)
Jobless	
Yes	645 (78.37)
Financial resources	
No resource	413 (50.18)
<500 Euros	220 (26.73)
>500 et <1000 Euros	132 (16.04)
>à 1000 Euros	58 (7.05)
COVID	
Confirmed	56 (6.81)
COVID contact	
Yes	195 (23.69)
Food Consumption	
3 meals a day	429 (52.13)
1 meal per day	353 (42.89)
<than 1 meal a day	41 (4.98)
**Psychiatric symptoms**	
PHQ-9	9.35 [2.11]
GAD-7	6.37 [5.96]
IES-R	22.35 [5.96]
**Psychiatric symptoms with cut-offs**	
PHQ-9 ≥ 10	378 (45.93)
GAD-7 ≥ 8	302 (36.69)
IES-R ≥ 26	321 (39.00)
Suicidal ideation	209 (25.39)

Legend: Abbreviation: PHQ-9, 9-item Patient Health Questionnaire; GAD-7, 7-item Generalized Anxiety Disorder; IES-R, 22-item Impact of Event Scale-Revised.

**Table 2 nutrients-15-03260-t002:** Odds ratio and identification of risk for mental health by logistic regression.

	Depression (PHQ-9)			Anxiety (GAD-7)			Distress (IES-R)			Suicidal Ideation		
	Affected No. (%)	Odds Ratio (95%IC)	*p*-Value	Affected No. (%)	Odds Ratio (95%IC)	*p*-Value	Affected No. (%)	Odds Ratio (95%IC)	*p*-Value	Affected No. (%)	Odds Ratio (95%IC)	*p*-Value
**SEX**												
MALE	173 (45.77)	REFERENCE	NA	133 (44.04)	REFERENCE	NA	161 (50.16)	REFERENCE	NA	105 (50.24)	REFERENCE	NA
FEMALE	205 (54.23)	1.39 (1.05–1.83)	**0.02**	169 (55.96)	1.47 (1.11–1.96)	**0.007**	160 (49.84)	1 (0.76–1.33)	1	104 (49.76)	1 (0.73–1.37)	1
**LEVEL STUDY**												
HIGHER THAN BACCALAUREATE	59 (15.61)	REFERENCE	NA	50 (16.56)	REFERENCE	NA	47 (14.64)	REFERENCE	NA	26 (12.44)	REFERENCE	NA
BACCALAUREATE	140 (37.04)	0.95 (0.62–1.48)	0.09	114 (37.75)	0.90 (0.58–1.39)	0.66	108 (33.65)	0.91 (0.59–1.42)	0.74	71 (33.97)	1.17 (0.70–1.95)	0.61
LOWER THAN BACCALAUREATE	179 (47.35)	0.70 (0.46–1.06)	0.09	138 (45.69)	0.63 (0.41–0.96)	**0.04**	166 (51.71)	0.93 (0.62–1.42)	0.83	112 (53.59)	1.23 (0.76–2.01)	0.47
**HOUSING**												
FAMILY	236 (62.43)	REFERENCE	NA	182 (60.27)	REFERENCE	NA	186 (57.94)	REFERENCE	NA	108 (51.67)	REFERENCE	NA
TENANT	78 (20.64)	0.91 (0.65–1.29)	0.66	65 (21.52)	1.05 (0.73–1.49)	0.79	73 (22.74)	1.22 (0.86–1.73)	0.28	51 (24.40)	1.51 (1.02–2.22)	**0.04**
EMERGENCY SHELTER	31 (8.20)	1.07 (0.64–1.80)	0.79	25 (8.28)	1.14 (0.67–1.94)	0.68	31 (9.66)	1.60 (0.96–2.70)	0.08	22 (10.53)	1.92 (1.10–3.35)	**0.03**
HOMELESS	33 (8.73)	1.79 (0.70–1.97)	0.60	30 (9.93)	1.52 (0.90–2.54)	0.13	31 (9.66)	1.56 (0.93–2.61)	0.10	28 (13.40)	2.77 (1.62–4.71)	**<0.001**
**FINANCIAL RESOURCES**												
>1000 €UROS	22 (5.82)	REFERENCE	NA	21 (6.95)	REFERENCE	NA	12 (3.74)	REFERENCE	NA	8 (3.83)	REFERENCE	NA
>500 ET < 1000 €UROS	68 (17.99)	1.73 (0.92–3.26)	0.11	57 (18.87)	1.34 (0.71–2.53)	0.43	55 (17.13)	2.73 (1.33–5.64)	**0.005**	36 (17.22)	2.34 (1.01–5.42)	0.06
<500 €UROS	101 (26.72)	1.39 (0.77–2.51)	0.30	81 (26.83)	1.03 (0.56–1.87)	1	98 (30.53)	3.08 (1.55–6.13)	**<0.001**	61 (39.19)	2.39 (1.07–5.35)	**0.04**
NO RESOURCES	187 (49.47)	1.35 (0.77–2.38)	0.32	143 (47.35)	0.93 (0.53–1.65)	0.88	156 (48.60)	2.32 (1.29–4.53)	**0.01**	104 (49.76)	2.10 (0.96–4.58)	0.07
**LIVE ALONE**												
NO	249 (65.87)	REFERENCE	NA	195 (64.57)	REFERENCE	NA	206 (64.17)	REFERENCE	NA	121 (57.89)	REFERENCE	NA
YES	129 (34.13)	1.28 (0.95–1.72)	0.11	107 (35.43)	1.36 (1.00–1.84)	0.05	115 (35.83)	1.41 (1.05–1.91)	**0.025**	88 (42.11)	1.91 [1.38–2.65]	**<0.001**
**WORKING**												
YES	72 (19.05)	REFERENCE	NA	63 (20.86)	REFERENCE	NA	57 (17.76)	REFERENCE	NA	43 (20.57)	REFERENCE	NA
NO	306 80.95)	1.32 (0.95–1.86)	0.11	239 (79.14)	1.07 (0.76–1.52)	0.73	264 (82.24)	1.47 (1.03–2.09)	**0.04**	166 (79.43)	1.08 [0.74–1.60]	0.70
**COVID DIAGNOSIS**												
NO	344 (91.01)	REFERENCE	NA	272 (90.07)	REFERENCE	NA	292 (90.96)	REFERENCE	NA	194 (92.82)	REFERENCE	NA
YES	34 (8.99)	1.90 (1.09–3.31)	**0.025**	30 (9.93)	2.09 (1.21–3.62)	**0.009**	29 (9.04)	1.74 (0.97–3.13)	**0.047**	15 (7.18)	1.08 (0.58–1.99)	0.87
**COVID CONTACT**												
NO	266 (70.37)	REFERENCE	NA	216 (71.52)	REFERENCE	NA	228 (71.03)	REFERENCE	NA	162 (77.51)	REFERENCE	NA
YES	112 (29.63)	1.83 (1.32–2.54)	**<0.001**	86 (28.48)	1.50 (1.08–2.09)	**0.02**	93 (28.97)	1.83 (1.32–2.54)	**<0.001**	47 (22.49)	0.91 (0.63–1.33)	0.71
**FOOD INSECURITY**												
2 AND MORE MEALS	163 (43.12)	REFERENCE	NA	134 (44.36)	REFERENCE	NA	140 (43.62)	REFERENCE	NA	77 (36.84)	REFERENCE	NA
1 MEAL	193 (51.06)	1.97 (1.48–2.62)	**<0.001**	148 (49.01)	1.59 (1.18–2.13)	**0.002**	160 (49.84)	1.71 (1.27–2.29)	**<0.001**	111 (53.11)	2.1 (1.50–2.93)	**<0.001**
<1 MEAL	22 (5.82)	1.89 (0.99–3.59)	0.07	22 (7.28)	2.10 (1.10–4.00)	**0.04**	21 (6.54)	2.16 (1.14–4.13)	**0.02**	21 (10.05)	4.8 (2.48–9.29)	**<0.001**

Legend: Abbreviation: PHQ-9, 9-item Patient Health Questionnaire; GAD-7, 7-item Generalized Anxiety Disorder; IES-R, 22-item Impact of Event Scale-Revised. Boldface indicates the significance of variables.

**Table 3 nutrients-15-03260-t003:** Adjusted Logistic Regression Analysis of food insecurity associated with depression, anxiety, distress and suicidal ideation.

	Depression (PHQ-9)			Anxiety (GAD-7)			Distress (IES-R)			Suicidal Ideation		
	Affected No. (%)	Odds-Ratio (95%IC)	*p*-Value	Affected No. (%)	Odds-Ratio (95%IC)	*p*-Value	Affected No. (%)	Odds-Ratio (95%IC)	*p*-Value	Affected No. (%)	Odds-Ratio (95%IC)	*p*-Value
**FOOD INSECURITY**												
2 AND MORE MEALS	163 (43.12)	REFERENCE	NA	134 (44.36)	REFERENCE	NA	140 (43.62)	REFERENCE	NA	77 (36.84)	REFERENCE	NA
1 MEAL	193 (51.06)	2.11 (1.57–2.84)	**<0.001**	148 (49.01)	1.70 (1.25–2.30)	**0.001**	160 (49.84)	1.71 (1.27–2.30)	**0.001**	111 (53.11)	2.10 (1.50–2.96)	**<0.001**
<1 MEAL	22 (5.82)	2.30 (1.19–4.51)	**0.005**	22 (7.28)	2.51 (1.29–4.88)	**0.006**	21 (6.54)	2.36 (1.23–4.57)	**0.01**	21 (10.05)	4.81 (2.46–9.44)	**<0.001**

Legend: Abbreviation: PHQ-9, 9-item Patient Health Questionnaire; GAD-7, 7-item Generalized Anxiety Disorder; IES-R, 22-item Impact of Event Scale-Revised. Boldface indicates the significance of variables

## Data Availability

The data can be transmitted by the corresponding author.

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
