# Peer review of "Precarious Young Adults’ Mental Health during the Pandemic: The Major Impact of Food Insecurity Independently of COVID-19 Diagnosis"

_nutrients, 2023, doi:10.3390/nu15143260_

Round 1

Reviewer 1 Report

After reading the article, I noticed the following aspects related to:

Abstract. The authors clearly specify the main objective of the research, the research methodology carried out, the results and conclusions of the research.

Introduction. The authors specify two research objectives. I suggest the authors specify which gap is covered by the specialized literature and the way of describing the sections in the article. I also do not find the literature review part distinct or separate. For this reason, this part is deficient and I suggest enriching this section.

Methods. The authors clearly describe the methodology used, the data measurement method and their statistical analysis.

Results. The authors describe the data obtained by transposing them into tables.

Discussion. The authors describe and interpret the results, specify the limits of the study.

Conclusions. The authors describe their own research contributions.

No comments.

Author Response

We would like to thank the reviewer for his/her help to improve our manuscript. We are very grateful. For ease of review, changes have been in green in the text.

Point1. In line with the reviewer's comments, we have had the manuscript proofread to improve the quality of the English. The changes are shown in the text following the modifications.

Point2. Reviewer comment: Introduction. The authors specify two research objectives. I suggest the authors specify which gap is covered by the specialized literature and the way of describing the sections in the article. I also do not find the literature review part distinct or separate. For this reason, this part is deficient and I suggest enriching this section.

Response: We would like to thank the editor for his advice. We have added the following paragraph at the end of the introduction:

However, few studies specifically address the mental health of precarious youth during the pandemic. Indeed, the French literature has shown little interest in the issue of food insecurity during the pandemic. A study on the homeless population indicated the importance of food distress on their mental health [4], and a study on the general population also found a reduction in the PNNS-GS2 index among students experiencing financial precariousness, food insecurity, and psychological distress [21]. However, we have not identified any studies on this topic among a population of young precarious individuals who are not enrolled in university. Consequently, it can be affirmed that there is a lack of research on this population.

Reviewer 2 Report

This cross-sectional study analyzed the mental health of 823 French precarious young people, trying to find out an association with food insecurity, in the context of the COVID-19 pandemic and the consequent lockdown. Food insecurity resulted associated with depression, anxiety, distress and suicidal ideation.

The various sections of the manuscript are well organized and described. Nevertheless, it should be mentioned that food insecurity causes malnutrition, and that malnutrition can adversely affect mental health. Furthermore, poor mental health may be the cause of choosing to lead a precarious life. These aspects should be briefly discussed and added to the limits of the study.

Author Response

We would like to thank the reviewer for his/her help to improve our manuscript. We are very grateful. For ease of review, changes have been in green in the text.

Point 1.: Nevertheless, it should be mentioned that food insecurity causes malnutrition, and that malnutrition can adversely affect mental health.

Response 1.: We thank the reviewer for his comments. A sentence has been added to page 8, line 306 - 309. Reference 39 completes this element.

It is also important to highlight that food insecurity leads to malnutrition, and that mal-nutrition can have detrimental effects on mental health [39].

Point2.: Furthermore, poor mental health may be the cause of choosing to lead a precarious life.

Response 2.: We thank the reviewer for his comments. We've added the following sentence to the limit section.

It could also have shed light on the bi-directionality between precariousness and mental health.